# Economic and Environmental Potential of Wire-Arc Additive Manufacturing

**Manuel Dias, João P. M. Pragana** **, Bruna Ferreira, Inês Ribeiro** and **Carlos M. A. Silva ***

IDMEC, Instituto Superior Técnico, Universidade de Lisboa, Av. Rovisco Pais, 1049-001 Lisboa, Portugal;
manuel.e.dias@tecnico.ulisboa.pt (M.D.); joao.pragana@tecnico.ulisboa.pt (J.P.M.P.);
bruna.ferreira@tecnico.ulisboa.pt (B.F.); ines.ribeiro@ist.utl.pt (I.R.)
* Correspondence: carlos.alves.silva@tecnico.ulisboa.pt

**Abstract:** Since its creation, Additive Manufacturing (AM) has experienced a tremendous growth particularly over the last decade due to the industrial paradigm shift intended for improving conventional manufacturing procedures. This work is focused on an emerging AM process known as Wire-Arc Additive Manufacturing (*WAAM*) to assess its potential for further applications involving metallic costumer-oriented parts. Contrary to most AM processes, *WAAM* allows deposition of material layer-by-layer to be accomplished under high deposition rates, low production times and near 100% material efficiency using accessible equipment. The work stems from evaluating the economic viability in the production of parts by *WAAM* as an alternative for conventional processes such as those used in traditional subtractive approaches. For that purpose, a process-based cost model (PBCM) was developed for estimating production costs using a strong technological approach. The PBCM was tested with the production of a case study part by *WAAM* and its environmental impact was further assessed through life cycle assessment (LCA). Results show that *WAAM* can be economically and environmentally viable within specific industrial contexts. Moreover, further developments and optimizations of process variables and equipment will allow this technology to mature into tackling novel production challenges in a time and cost-effective manner.

**Keywords:** wire-arc additive manufacturing; process-based cost model; life cycle assessment; case study

## 1. Introduction

The trend of mass customization and the need in industry for producing lightweight parts of increasingly complexity in terms of overall shape or tailor-designed features is challenging the manufacturing industry in significantly pushing production chains to their limits in a time known as the 4th Industrial Revolution [1]. In this view, the need for providing products and/or services that best fit consumption needs while maintaining near mass production efficiency is of great importance [2].

In view of the above, one technology that is nowadays standing out as a key enabler for flexible production of tailor-made end-use components with sophisticated shapes/features is Additive Manufacturing (AM) [3]. Although originally used to produce prototypes, AM it is nowadays utilized to produce fully dense parts for state-of-the-art applications in a wide variety of materials ranging from plastics, organics, ceramics and composites to metals [4]. In case of metals, the most widespread AM processes belong to the categories of Powder Bed Fusion (PBF) and Direct Energy Deposition (DED) [5].

PBF processes work by selectively melting several beds of metallic powder placed over a platform layer-by-layer for shaping the final part. These processes use focused thermal heat sources in form of lasers or electron beams to allow printing complex parts with high resolution. However, PBF is largely affected by drawbacks associated to expensive energy and raw material consumptions, complex and costly equipment, limited envelops and

to the necessity of using support structures that can be arduous or even impossible to remove [6].

DED processes make use of feedstock (which can be powder or wire) that is fed and melted instantaneously with the combined action of a thermal heat source and a feedstock supply unit. Moreover, DED offers the possibility of utilizing an alternative and much cheaper heat source in the form of an electric arc. This variant is designated as wire-arc additive manufacturing (*WAAM*) and has been attracting some attention over the years from the manufacturing sector due to its accessible and sustainable use to fabricate large scale parts with high deposition rates, low-cost equipment and high material efficiency [7]. In addition, the acquisition costs of *WAAM* equipment may even be negligible since most companies already have at their disposal the necessary electric arc welding and computer numeric control (*CNC*) motion systems [8].

Nowadays, the applicability of *WAAM* is rising in accordance to the increasingly worldwide profits of metal AM [9]. Examples of such can be found in parts assembled in the jet airline model Boeing 787 Dreamliner [10], structural components such as full bridges made by MX3D [11], pressure vessels for space exploration by Thales Alenia Space [12], among others. However, *WAAM* is still in its infant stage in terms of state-of-the-art implications concerning deposition strategies, parametrization, part quality or new materials, all of which are currently under extensive investigation [13]. Moreover, the economic and environmental performance of *WAAM* are key points in forecasting and settling on further applications of this technology as an efficient and sustainable alternative to conventional manufacturing technologies.

In terms of cost evaluation, some studies have been published in recent years surrounding AM processes with comparisons to other manufacturing technologies such as [14–16] addressing several AM processes such as Direct Metal Laser Sintering, Electron Beam Melting, Stereolithography, Fused Deposition Modeling, among others [17]. Regarding *WAAM*, the first cost model, to the authors concern, was developed by Martina and Williams [18] to compare the cost of a titanium part produced by *WAAM* or by a conventional manufacturing chain of machining from solid. The overall cost modelling was time activity based for simplifying deposition times as a function of the CAD model volume and the buy-to-fly (*BTF*) ratio to estimate the specific cost of each process variable. The authors concluded that *WAAM* can achieve cost savings in the range of 7% to 69%. However, this cost model is not robust because it excessively simplifies the operator actions by considering only one operator per machine while not also accounting for important non-production tasks such as consumable changeovers.

More recently, another *WAAM* cost model was proposed by Cunningham et al. that particularizes in studying the main process activities apiece by sensitivity analysis [19]. This cost model is more robust than the previous developed by Martina and Williams because it considers other process chain variables, such as those involving baseplate preparation, part inspection or post-processing by heat treatment. However, the cost model does not consider dynamic on-off modifications such as those concerning consumables, overhead costs, different materials and variable labor.

Adding to the necessity of designing more complete and robust cost models for *WAAM*, the available literature on this technology still lacks a clear combined-effect assessment of cost and environmental impact that are crucial to ensure a proper process planning and design [20].

Under these circumstances, this work aims to investigate on the economic and environmental benefits that *WAAM* may bring in the near future. For this purpose, the state-of-the-art developments mentioned above will be extended with the establishment of a comprehensive process-based cost model (PBCM) for *WAAM*. In addition to metal deposition activities, the proposed model englobes other manufacturing stages that are intrinsic to a typical *WAAM* processing chain, such as machining operations, and allows inputting labor, materials, energy, equipment and building variables for the different manufacturing stages. Afterwards, the PBCM is tested with the fabrication of a case study

part and the results are compared against those obtained using a traditional subtractive approach (machining). The environmental impact will be estimated by means of Life Cycle Assessment (LCA) in the ReCiPe midpoint and endpoint levels. All in all, the proposed PBCM for a *WAAM* chain improves on previous ones in terms of robustness and capability of reacting to industry market dynamics. The cost performance results combined with those from LCA highlight the economic viability and environmental friendliness of *WAAM* when compared with conventional manufacturing technologies.

## 2. Methodologies

### 2.1. WAAM Processing Chain

The working principles of *WAAM* involve layer-by-layer deposition in which an electric arc is used as thermal energy source for melting the wire feedstock [21]. However, although metal deposition constitutes the core stage of the process, a typical *WAAM* production chain possesses other stages that are crucial for ensuring proper quality standards of metal parts. The best example of such corresponds to the utilization of machining operations because quality of as-built parts produced by *WAAM* can be very rough in terms of surface quality and geometric precision which are far from conventional quality standards of end-use metal parts [22].

Figure 1 presents a flow map of a typical *WAAM* processing chain which can be decomposed in five stages: (1) pre-processing, (2) equipment setup, (3) metal deposition by *WAAM*; (4) machining and (5) post-processing.

The pre-processing stage (1) concerns all tasks involved in modelling and planning the following four stages. Examples of these tasks are 3D part modelling, choosing the adequate processing parameters or designing the deposition strategy to be used during stage (3).

The equipment setup stage (2) is implemented in-site and involves setting up the machines, preparing and clamping the baseplate and installing the process consumables such as wire feedstock and shielding gas. In a typical *WAAM* processing chain, three main machines are needed: (i) the power unit that feeds and simultaneously melts the wire feedstock, (ii) the motion system to move the torch in pre-defined regions for depositing metal and (iii) the machining center to be used in stage (4) of the chain.

The metal deposition stage (3) is where a near-net-shape part is built according to the strategies and processing parameters established in the previous stages.

The machining stage (4) is where the near-net-shape part will be shaped into one with the required geometric precision, dimensional tolerances, and surface quality. Additionally, this stage may also account for other cutting tasks such as those used for sawing the deposited part from the baseplate if needed.

Finally, a *WAAM* processing chain reaches its end with a post-processing stage (5). This stage does not include any machining operations but instead other closure tasks such as heat treatments, polishing, part inspection, painting, transportation, among others.

From all five stages schematized in Figure 1, stages (2), (3) and (4) are the most exclusive ones of the AM technology with special regard to *WAAM*. In fact, stages (1) and (5) involving pre and post processing are also customary in conventional manufacturing technologies such as machining from a solid or casting. Moreover, the duration and strictness level of both these stages are highly dependent on not only the part itself but also on its intended industrial applicability context. For these reasons, stages (1) and (5) are not included in the proposed PBCM which in turn will enable a more suitable and fair comparison between *WAAM* and conventional manufacturing technologies.

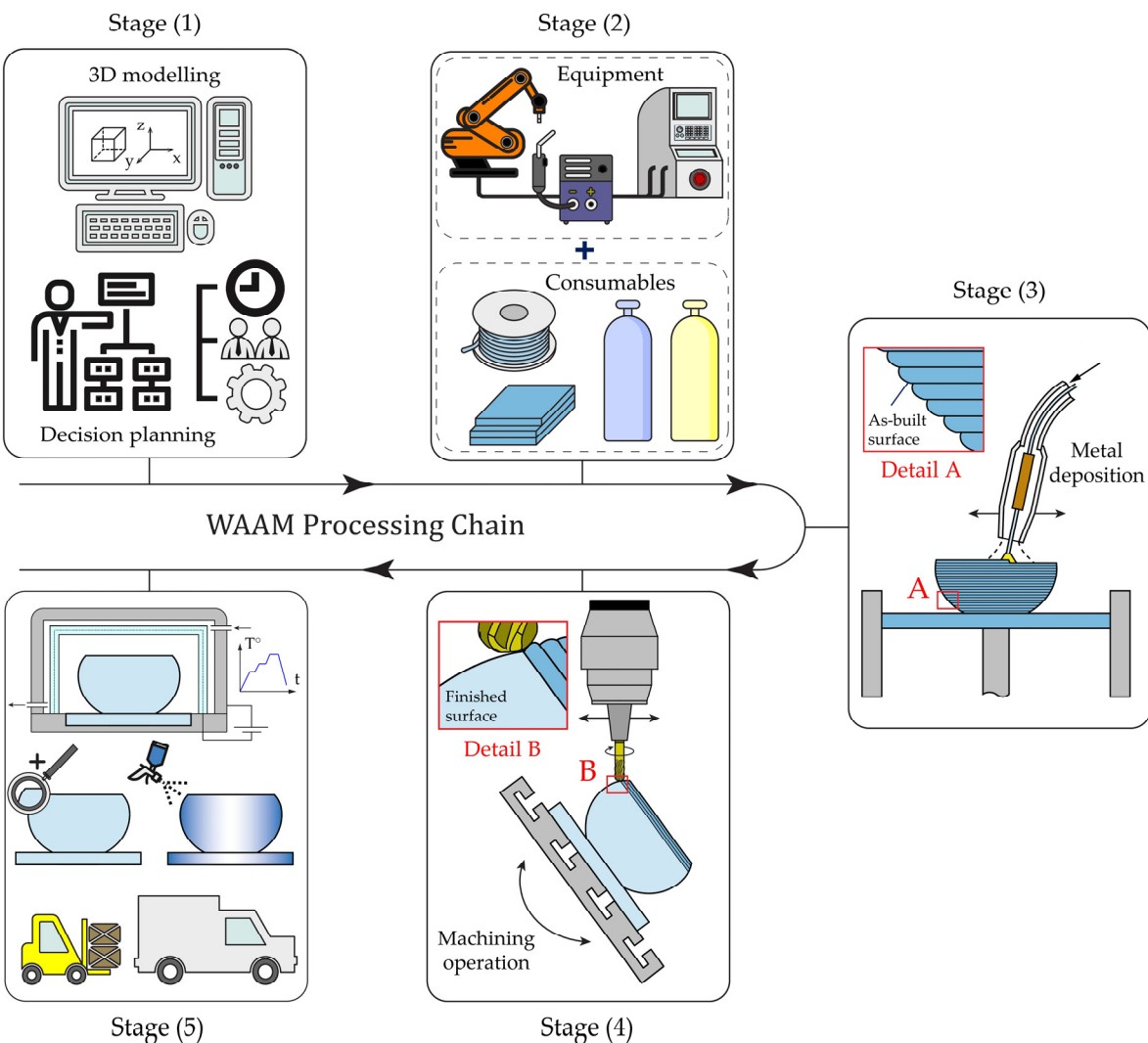

**Figure 1.** Flow map of a typical *WAAM* processing chain composed by five main stages.

### 2.2. Cost Model Analysis

The development of a PBCM for the main three stages of a *WAAM* processing chain (refer to stages (2), (3) and (4) shown in Figure 1) is aimed at fulfilling the following three main aspects: (i) to establish a strong relation between process, resources, product, equipment and other cost-based variables; (ii) to ensure its simple and accessible reproducibility within different scenarios and (iii) to be capable of reacting to sudden or predicted changes in the overall workflow.

The PBCM will be mostly based on the pre-defined annual production that allows outputting the total annual costs $C_{total}$ which are estimated as the sum of fixed costs $C_{fixed}$ with variable costs $C_{variable}$,

$$C_{total} = C_{fixed} + C_{variable} \tag{1}$$

#### 2.2.1. Fixed Costs

The fixed costs included in the PBCM are those that are not directly associated with the annual production of goods. These costs do not change during the production cycle and can be obtained by summing up cost activities associated to equipment, building use and administration overheads,

$$C_{fixed} = C_{equipment} + C_{building} + C_{overhead} \tag{2}$$

The equipment costs $C_{equipment}$ englobe expenses on machines, maintenance, devices, fixtures and tools, which can be further divided into the following three terms,

$$C_{equipment} = C_{machinery} + C_{tooling} + C_{maintenance} \qquad (3)$$

The first term refers to the acquisition costs of the three main machines utilized in a typical *WAAM* processing chain: an electric arc welding power source, a motion system and a *CNC* machining center. These investments are dependent on the cost of opportunity and years of life $N_{life}$ associated with each machine as well as to their usability as dedicated or non-dedicated equipment. In cases where the machines are non-dedicated, their investments $M_{machine}$ must be multiplied by their utilization rate $UR$ as follows,

$$C_{machinery} = \frac{M_{machine}}{N_{life}} \times UR = \frac{M_{machine}}{N_{life}} \times \frac{t_c \times AP \times (1 + RR/100)}{t_u} \qquad (4)$$

where $t_c$ is the total cycle time, AP is the annual production (number of parts per year), RR is the rejection rate (%) and $t_u$ is the machine uptime given by the multiplication of the annual working days with the machine daily utility time. Equation (4) is computed for both metal deposition and machining stages separately by inserting their respective machine investments and corresponding cycle times $t_c = t_{WAAM}$ and $t_c = t_{machining}$. The acquisition of special-purpose devices such as torches or alignment devices, among others, should be included in the first term of Equation (3).

The second term of Equation (3) is for accounting costs associated with the tools needed in the main stages of the *WAAM* processing chain. In this parcel, the costs will be mainly dependent on the acquisition of machining tools due to their limited tool lifetime [Park, 2020] and can be estimated as follows,

$$C_{tooling} = \sum_{i=1}^{n} \left( \frac{n \times C_{cutting} \times t_{cutting}}{t_l} \times UR \right)_i + C_{fixing} \qquad (5)$$

where $n$ is the number of cutting tools, $C_{cutting}$ is the acquisition cost for each cutting tool, $t_c$ is the actual cutting time and $t_l$ is the tool lifetime. Additionally, tooling costs can also include those spent on tool holders and fixtures or jigs used for clamping the workpiece for both metal deposition and machining tasks ($C_{fixing}$ in Equation (5)).

The third and final term of Equation (3) allocates expenses related to annual maintenance activities on the three main machines used in a *WAAM* chain. These include cleaning, lubrification and replacement of worn-off accessories such as torch contact tips and nozzles for maintaining the mechanical assets of the machines.

The remaining two terms involved in the fixed costs (refer to Equation (2)) are related to building use and administration overheads. Building use expenses $C_{building}$ are estimated as a function of the yearly rent rate (YRR, €/m$^2$), the machinery area (MA, m$^2$) and the utilization rate (UR),

$$C_{building} = YRR \times MA \times UR \qquad (6)$$

The overheads costs $C_{overhead}$ are not directly spent on the processing chain but are allocated to the industrial/fixed structure, not directly related with the production line.

### 2.2.2. Variable Costs

The variable costs are those that show a direct dependence on the annual production of the desired goods. These costs correspond to the sum of expenses associated with the following three main groups: materials, labor and energy,

$$C_{variable} = (C_{mat} - I_{waste}) + C_{labor} + C_{energy} \qquad (7)$$

The term $C_{mat}$ accounts for metals and consumables such as wire feedstock, baseplate and shielding gas needed for carrying out the main stages of the production chain. These

costs are presented per unit while also accounting for material expenses spent on rejected parts with the inclusion of the rejection rate RR (%),

$$C_{mat} = \left[ \frac{C_w \times \rho_w}{m_w} (V_{as-built}) + \frac{C_b \times \rho_b}{m_b} \left( \frac{V_b^{eff}}{N_b} \right) + \frac{C_{gas} \times Q_{gas}}{V_{gas}} \left( t_{dep} \right) \right] \times \left( 1 + \frac{RR}{100} \right) \quad (8)$$

The first term of Equation (8) englobes costs associated with the wire feedstock. This consumable is mostly bought in coils (similarly to electric arc welding applications) with a given unit cost $C_w$ and mass $m_w$ related to the density $\rho_w$ of the chosen metal alloy. Differences in part volume after deposition (as-built part) and after machining (final part) are conveyed by the Buy-to-Fly ratio (*BTF*) which corresponds to the ratio between the as-built part volume and the final part volume. As seen, the proposed PBCM allows inputting variables for different metal alloys given their corresponding material properties and unit acquisition cost, where the latter can vary from company to company.

The second term accounts all costs related to the baseplate material. This term is expressed in a similar way to that of the wire feedstock by being dependent of the acquisition unit cost $C_b$ and of its mass $m_b$ and density $\rho_b$. The effective baseplate volume $V_b^{eff}$ needed for supporting metal deposition of one part and its number of uses $N_b$ are also expressed. In cases where the baseplate is designed as a segment of the final part, it cannot be reusable (i.e., $N_b = 1$).

The third term is related to shielding gas consumables needed for protecting the molten metal from environmental contaminations during depositions. These consumables are usually acquired in gas cylinders or tanks with a given cost $C_{gas}$ and gas volume $V_{gas}$. Since this consumable is only being used when metal deposition is taking place, its expenses are dependent on the gas flow rate $Q_{gas}$ and deposition time $t_{dep}$, where the first is a processing parameter of *WAAM* and the second can be determined as a function of the wire feed speed *WFS* and diameter $\varnothing_w$ as follows,

$$t_{dep} = \frac{4}{\pi} \times \frac{1}{WFS \times \varnothing_w{}^2} \times \left( V_{part} \times BTF \right) \quad (9)$$

The term $I_{waste}$ of Equation (7) considers incomes from waste management resulting from metal deposition tasks or machining tasks to scrap processing companies,

$$I_{scrap} = \left[ V_{as-built} \left( \frac{RR_{WAAM}}{100} \right) + V_{part} \left( \frac{RR_{machining}}{100} + (BTF - 1) \right) \right] \times C_{Scrap} \quad (10)$$

As seen, these incomes will be highly dependent on the waste income per volume $C_{Scrap}$ and include all solid wastes in the form of rejected parts for both manufacturing stages $RR_{WAAM}$ and $RR_{machining}$ as well as metal chips that come from machining stages by means of the parameter $(BTF - 1)$ multiplied by the part volume $V_{part}$.

The labor costs were considered only for operators directly involved in the different stages of a *WAAM* processing chain. These costs are estimated as follows,

$$C_{labor} = M_{Labor} \times \left( t_{WAAM} + t_{machining} + t_{setup} \right) \quad (11)$$

where $M_{Labor}$ corresponds to annual direct wages of the total number of operators directly involved in the production chain. The variables inside the brackets correspond to different time-consuming stages that will be disclosed apiece.

The first input $t_{WAAM}$ is given by the sum of the deposition time $t_{dep}$ with the cooling time $t_{cooling}$ as follows,

$$t_{WAAM} = \left( t_{dep} + t_{cooling} \right) \times WD_{WAAM} \quad (12)$$

The cooling time (also known as dwell time) englobes all idle times taking place in between the deposition of material layers. The rightmost coefficient *WD* corresponds to the worker dedication percentage (in this case, during metal deposition by *WAAM*).

The second input $t_{machining}$ is the total machining time which considers two main components: the actual cutting time and idle times $t_{off}$. The first component implies contact between the workpiece and the cutting tool, while the other compiles all non-productive times such as those used for tool changeovers. This allows retrieving the following equation,

$$t_{machining} = \left( \frac{V_{part}\,(BTF - 1)}{MRR} + t_{off} \right) \times WD_{machining} \tag{13}$$

where *MRR* is the material removal rate and $t_{off}$ accounts for all non-productive times.

The third and final input of Equation (11) regards setup tasks involved in the operation of the *WAAM* and machining equipment and can be calculated with the following equation,

$$t_{setup} = \frac{t_{set\ WAAM} + t_{set\ machining}}{Parts\ per\ clamping} + t_{w\ change} \left( \frac{\rho_w}{m_w} \right) \times (V_{part} \times BTF) + t_{gas\ change} \times \left( \frac{V_{gas}/Q_{gas}}{t_{dep}} \right)^{-1} \tag{14}$$

As seen, the overall setup time $t_{setup}$ is calculated per part and is affected by the time it takes to set up both *WAAM* $t_{set\ WAAM}$ and machining $t_{set\ machining}$ equipment (inputting and compiling the *CNC* programs, machine offsetting, workpiece clamping, and other minor tasks). The term "Parts per clamping" is to account for the number of parts that can be fabricated in a single baseplate clamping (i.e., within the same setup time). Changeover times for replacing the wire coils $t_{w\ change}$ and the gas bottle/tank $t_{gas\ change}$ with new ones are also considered.

The expenses related to energy consumptions needed for operating the main machines of the *WAAM* production chain are obtained from the multiplication of the main power consumptions in *WAAM* and machining operations ($P_{WAAM}$ and $P_{machining}$, respectively) with their consumed time and with the monetary price of energy $M_{energy}$,

$$C_{energy} = \left[ \sum \left( P_{WAAM} \times t_{dep} \right) + \sum \left( P_{machining} \times t_{machining} \right) \right] \times M_{energy} \tag{15}$$

Since the main *WAAM* power consumption (welding power source + motion system) comes from the instants where metal deposition is taking place, only the deposition time is considered. For machining operations, the total machining time is used because the machining center is not only fully operating during the actual cutting time but also during tool changing times and idle times as well.

### 2.3. Life Cycle Assessment

The evaluation of the environmental aspects and potential impacts associated with the implementation and utilization of the *WAAM* processing route was developed following the Life Cycle Assessment (LCA) methodology. Although being a life cycle methodology, allowing for cradle to grave analyses of products (Ribeiro, 2020), this tool allows estimating the environmental performance of a process by adopting a cradle to gate scope, that is, analyzing only the raw material extraction and manufacturing phases.

The net of resources to be considered in the LCA of *WAAM* comprise the wire feedstock material, the baseplate material, the shielding gas and the sources of the electricity generation with resources and emissions taken from the Ecoinvent 3.7 database representing global averages. The environmental impact is quantified and later analyzed into different impact categories using the ReCiPe2016 method [23] in SimaPro 7 software developed by PRé Consultants, based in the Netherlands. Firstly, the ReCiPe Midpoint (H) V1.11/World Recipe H is used for transforming and evaluating the list of life cycle inventory under midpoint indicators among the chosen environmental impact categories to be presented.

Afterwards, the ReCiPe Endpoint (H) V1.11/World ReCiPe H/H method is also used for assessing the impacts into three main indicators: human health, ecosystems and resources.

## 3. Case Study

This section is structured in four different studies that make use of the established PBCM for evaluating the economic and environmental performance of *WAAM*. The starting point for these studies was on the production of a case study part schematized and illustrated in Figure 2a,b. The main processing parameters used for building the case study part were established in a previous work from the authors [24]. Since the proposed PBCM is based upon a volume-dependent approach, the reachable geometric complexity of as-built parts implies knowing beforehand on the most suitable deposition strategies by means of parametrization studies.

The first study is focused on presenting total production costs in *WAAM* for the main purpose of pointing out its main cost drivers. The second study comprises a sensitivity analysis to allow identifying the key fluctuations in production costs lead by modifications in the PBCM inputs. The third study is on assessing the economic viability of *WAAM* by comparing the proposed PBCM results with those obtained from a traditional subtractive approach by machining from solid. The fourth and final study is on analyzing the environmental viability of *WAAM* through LCA to allow concluding on its sustainability.

The case study part was firstly deposited by *WAAM* in a three-axis *CNC* gantry equipped with the gas metal arc welding power source LUC 400 Aristo 400 (from ESAB, Fulton, MD, USA). The materials used were AISI 316L stainless steel wire feedstock with 1 mm diameter, AISI 316L stainless steel hot-rolled baseplates with 15 mm thickness and a high-purity (99.99%) Argon gas for shielding purposes. The mechanical and metallurgical properties of AISI 316L stainless steel obtained by *WAAM* can be found elsewhere [25].

Machining tasks were carried out in the five axis universal milling machine DMU 50 incorporated with a rigid swivel rotary table for finishing the part in a single workpiece clamping within a machining time of 30 min. The total cycle times for each manufacturing stage were of $t_{WAAM} = 49$ min and $t_{machining} = 30$ min.

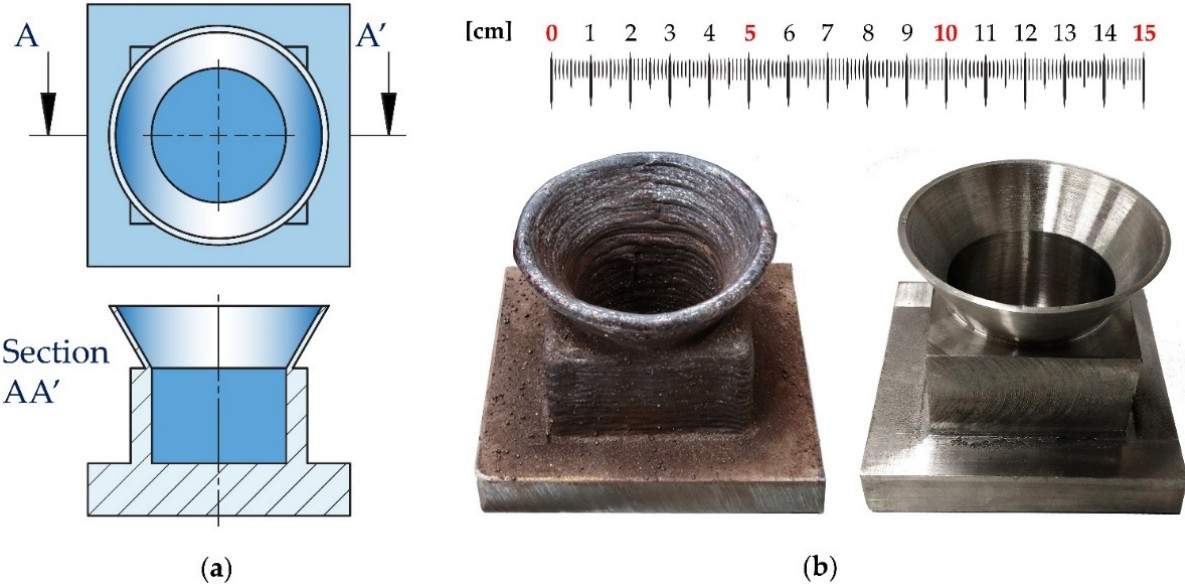

(**a**)        (**b**)

**Figure 2.** Case study part used for assessing the performance of the process-based cost model (PBCM). (**a**) Schematic representations on the top and front sectioned views of the part. (**b**) Case study part after deposition (**left**-side) and after machining (**right**-side).

Times spent on metal deposition by *WAAM* $t_{WAAM}$ and on machining $t_{machining}$ were measured onsite. Power consumptions in all three-phase three-wire circuits were assessed

with the power meter PROVA 6830 equipped with clamp-on ammeters and voltage test leads. The consumptions were recorded in 2 s time intervals with a resolution of 0.1 W and 1 W for power supplies up to 1 kW and 10 kW, respectively (for a maximum of 100 A). Figure 3 plots the experimental measurements for power consumptions from the gas metal arc welding source and *CNC* gantry during metal deposition of two layers by *WAAM*. These measurements allow concluding that: (i) power consumptions are high and nearly constant during metal deposition and (ii) power consumptions taking place during cooling times are negligible.

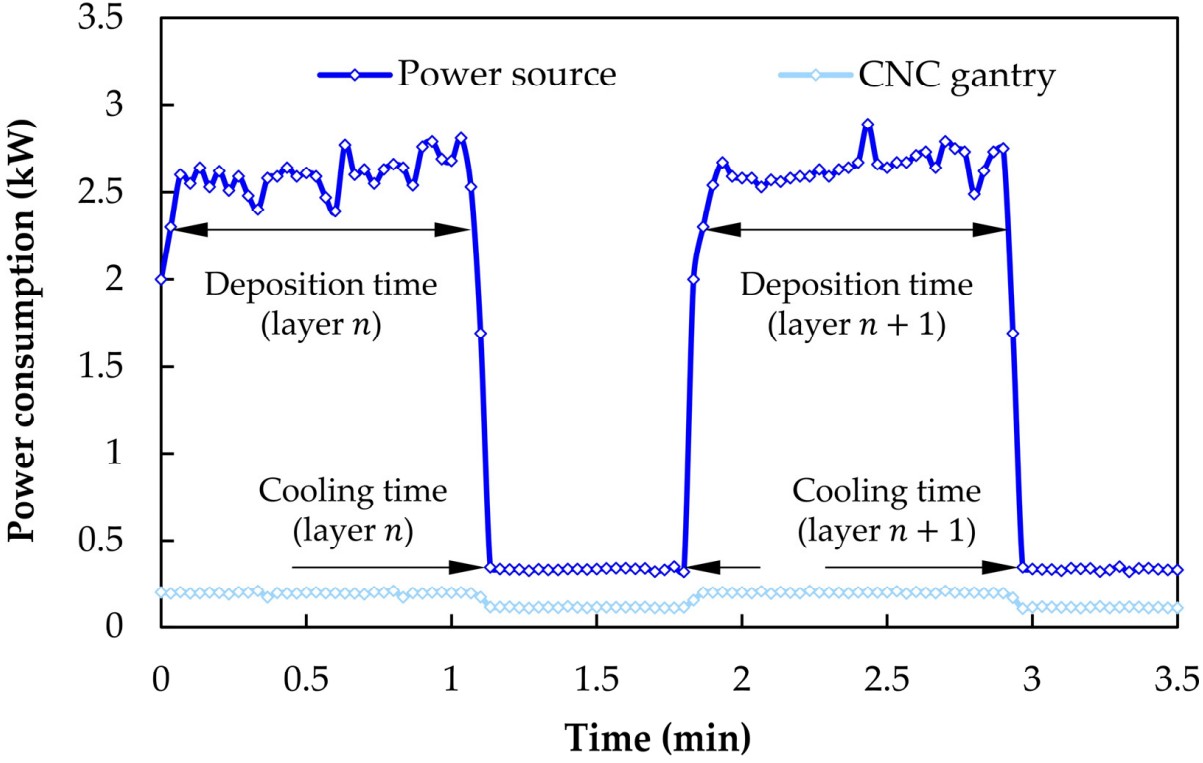

**Figure 3.** Experimental measurements of power consumptions (kW) from the gas metal arc welding power source (dark blue) and *CNC* gantry (light blue) during metal deposition by *WAAM* of two layers (layer "*n*" and layer "*n* + 1").

The assumptions considered in the PBCM that will be used in the different studies of Section 3 are disclosed in Table 1.

### 3.1. WAAM Cost Drivers

The cost performance of the *WAAM* processing chain will firstly be assessed by analyzing the production costs (€ per part) for metal deposition tasks. These costs are presented in Figure 4a in individual (left bar) and total (right bar) perspectives. The main conclusion drawn from Figure 4a is that production costs associated with metal deposition by *WAAM* are highly affected by material expenses. For this reason, the variable costs (23.20 € per part) clearly surpass the remaining fixed costs related to equipment and building use (2.64 € per part, corresponding to 10% of the total production costs). The sum between fixed and variable costs gives a total production cost of 25.84 € per part.

**Table 1.** Process-based cost modelling (PBCM) assumptions.

| Process-Independent Assumptions | Value |
| --- | --- |
| Annual production | 1500 parts |
| Yearly rent rate | 1500 €/m$^2$ |
| Machinery area | 25 m$^2$ |
| Machine uptime | 240 × 8 h/year |
| Machine life span | 5 years |
| Direct wages | 10 €/h |
| Monetary price of energy | 0.15 €/kWh |
| Yearly maintenance costs | 1700 € |
| Waste income (AISI 316L) | 0.0024 €/cm$^3$ |
| Administrative overheads | - |
| **_WAAM_ assumptions** | **Value** |
| Machine investment (power source + *CNC* gantry) | 20,000 € |
| Machine power consumption (during deposition) | 2.8 kW |
| Worker dedication | 5% |
| Machine setup time | 600 s |
| Wire cost per coil (15 kg) | 260 € |
| Wire density | 8 g/cm$^3$ |
| Wire feed speed | 6 m/min |
| Wire diameter | 1.0 mm |
| Wire changeover time | 300 s |
| Baseplate cost (20 cm × 50 cm × 2 cm) | 80 € |
| Effective baseplate volume (for 1 use) | 62 cm$^3$ |
| Baseplate density | 8 g/cm$^3$ |
| Gas cost (10,700 l) | 120 €/bottle |
| Gas flow rate | 10 L/min |
| Gas changeover time | 300 s |
| Total cooling time | 25 min |
| As-built part volume | 113 cm$^3$ |
| Rejection rate | 5% |
| **Machining assumptions** | **Value** |
| Machine investment | 100,000 € |
| Machine power consumption | 4.5 kW |
| Worker dedication | 20% |
| Machine setup time | 600 s |
| Number of tools | 4 |
| Tool cost per lifetime | 250 € |
| Material removal rate | 0.6 cm$^3$/min |
| Idle time | 2 min |
| Final part volume | 94 cm$^3$ |
| Rejection rate | 2% |

Figure 4b provides a deeper analysis on material costs showing that the wire feedstock contains a big portion (72%) of the overall material costs. The remaining costs spent on the baseplate and shielding gas are somewhat leveled, constituting percentages of 12% and 11% of the total material costs. These results are mainly related with the higher acquisition cost of the material in a wire format. Conversely, labor and energy costs are low due to the high automatization degree and high deposition rates of *WAAM* (low worker dedication and fast processing cycles) while equipment costs reflect upon the low acquisition expenses of flexible systems that integrate a welding power source with a *CNC* gantry.

Figure 5a shows the individual and total production costs (€ per part) for machining tasks that follow metal deposition in a *WAAM* processing chain. These costs are again presented in individual (left bar) and total (right bar) points of view and allow denoting that the equipment costs are now the main cost driver for machining the as-built parts. This leads to a complete shift from the previous analysis on metal deposition stages, whereby

the fixed costs are now the bigger portion of the total production costs (10.16 € per part corresponding to a percentage of 75%). The total machining costs are of 13.50 € per part which is nearly half of those spent on metal deposition by *WAAM* (25.84 € per part).

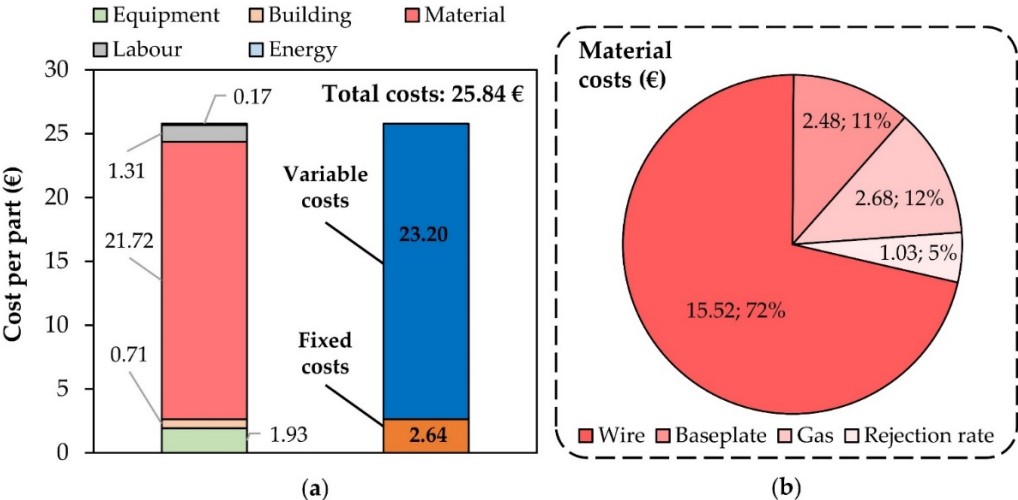

**Figure 4.** Cost performance of metal deposition stages in a *WAAM* processing chain. (**a**) Presentation of the individual and total production costs per part. (**b**) Presentation of the individual production costs per part related only to material costs.

Figure 5b explores the equipment expenses considered in the PBCM showing that they are mainly affected by machine costs (77%) due to high acquisition costs of the *CNC* machining center. Moreover, the higher machine size, worker dedication, and power consumption are responsible for the increases in building, labor and energy costs in machining tasks when compared with those obtained from metal deposition by *WAAM*. Since machining tasks are performed only on as-built parts, no material costs are included since they were fully accounted on the metal deposition tasks (refer to Figure 4).

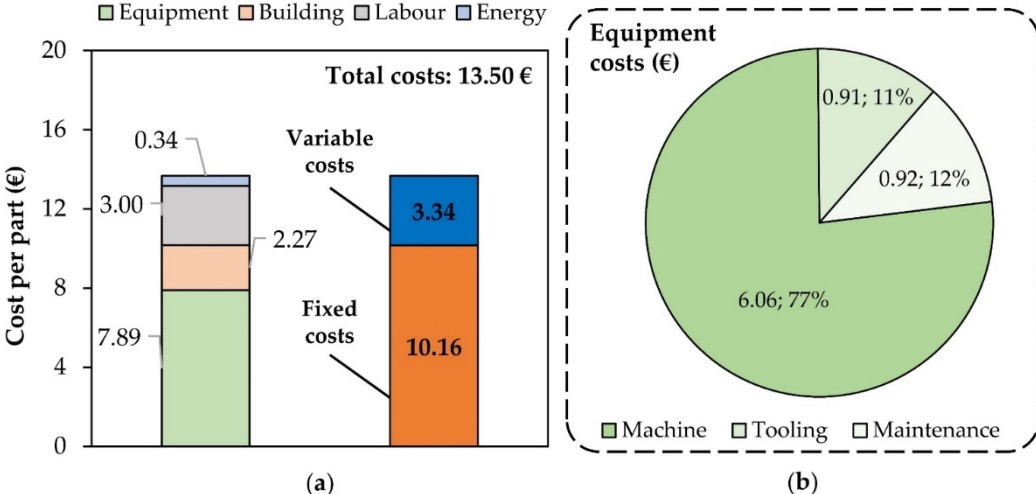

**Figure 5.** Cost performance of machining stages in a *WAAM* processing chain. (**a**) Presentation of the individual and total production costs per part. (**b**) Presentation of the individual production costs per part related only to equipment costs.

The complete distribution of costs per part for the *WAAM* processing chain is shown in Figure 6 for the individual and total perspectives. These distributions still evidence the preponderance of materials in *WAAM* (55%), equipment (25%) and labor (11%) as the main cost drives of the process. This analysis allows estimating a total production cost of 39.34 €

per part that can slightly decrease by considering solid waste incomes that are of 0.06 € per part.

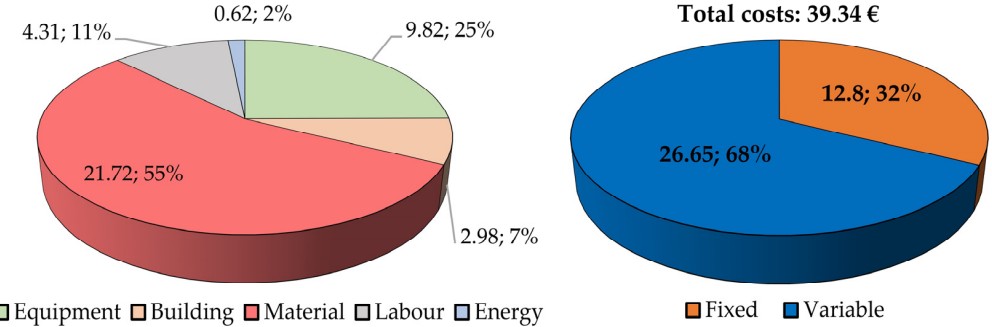

**Figure 6.** Cost performance of the complete *WAAM* processing chain with distributions of the individual (**left**) and total (**right**) costs per part (€).

### 3.2. Sensitivity Analysis

The sensitivity analysis is supported by the key cost drivers of the *WAAM* processing chain that were previously identified in Section 3.1. The goal here is to estimate the production cost fluctuations resulting from adjustments in the PBCM inputs as a consequence from industry market dynamics. For this purpose, the percentual variation of production costs per part is quantified by changing PBCM parameters within the range of ±30%.

Results are presented in Figure 7a for variable costs and Figure 7b for fixed costs. Variations in unit energy and labor costs show small fluctuations in the production costs, whereby the main cost driver is the setup time in machining tasks (maximum variation of ±3.47%). Still, this result highlights that setup times in machining have a greater impact in production costs than the same setup types but for metal deposition tasks (approximately 4 times higher fluctuations). As expected, the acquisition of the wire feedstock is the main cost drivers in a *WAAM* processing chain (maximum obtained variation of ±12.47%), clearly surpassing those spent in baseplate and shielding gas consumptions.

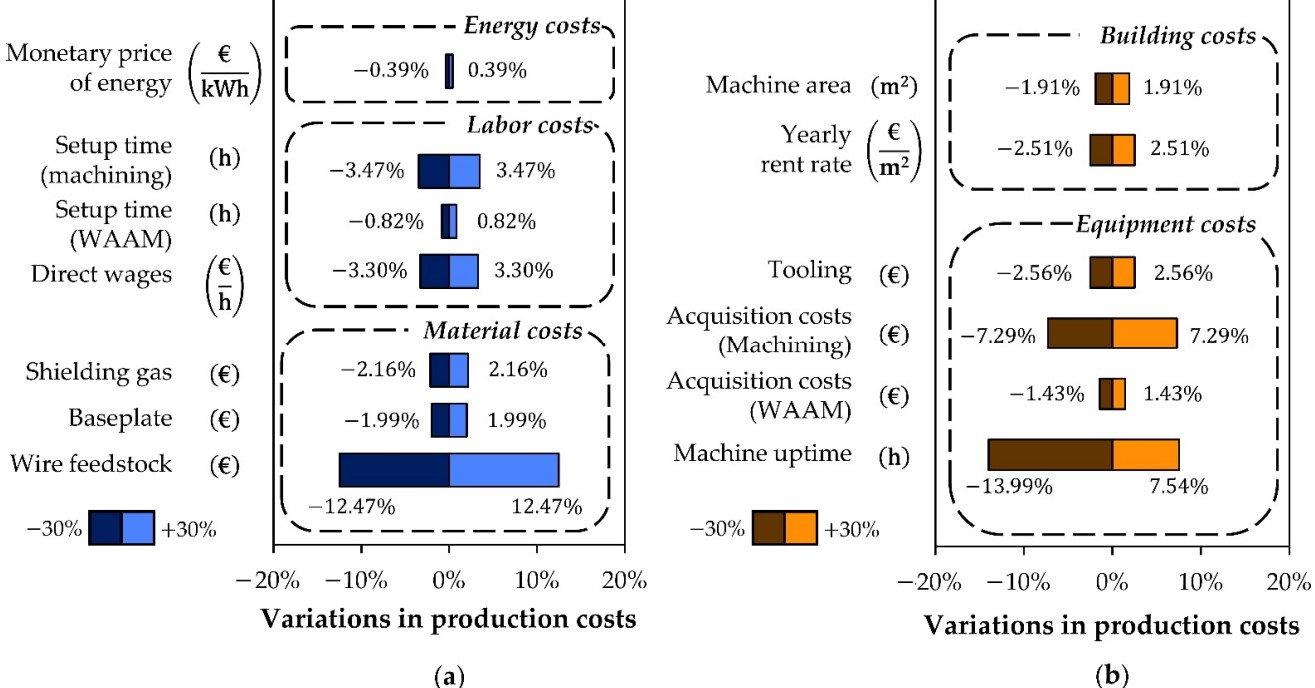

**Figure 7.** Sensitivity analysis on different PBCM inputs regarding the production cost per part. (**a**) Price fluctuations on variable costs. (**b**) Price fluctuations on fixed costs.

The fixed cost sensitivity analysis shown in Figure 7b allows denoting two main cost drivers: the acquisition costs of the *CNC* machining center and the machine uptime. The first is again related with the high acquisition cost of the machining equipment showing unit price fluctuations about 5 times higher than those attained for the metal deposition machines ($\pm 7.29\%$ against $\pm 1.43\%$, respectively). The machine uptime is related to its productive time (i.e., the time the equipment is working) and therefore its increase leads to negative fluctuations on production costs and vice versa. Moreover, fluctuations of the machine uptime originate asymmetric variations in production costs that show a higher absolute value ($-13.99\%$) when the machine uptime increases.

### 3.3. Comparison between WAAM and Machining from Solid

This subsection presents a comparison of the cost performance of a *WAAM* processing chain with that of a traditional subtractive approach upon which machining operations on billets or ingots give rise to the final part. For this purpose, the variables of the developed PBCM that are related with machining tasks were adapted for the traditional subtractive approach in two different manufacturing stages: a first one with a rough machining task followed by a fine machining task (Figure 8). These tasks are performed sequentially to allow decreasing machining times without compromising the final part quality standards in terms of surface conditions and geometry precision. The adaptations to be implemented in this Subsection made use of Equations (3)–(6), (13)–(15) of the proposed PBCM disclosed in Section 2.2.

For each task, two different Buy-to-Fly ratios (*BTF*) were considered: $BTF_1 = 3.2$ taken from a 500 cm³ raw stock and $BTF_2 = 1.2$ for fine finishing of the semi-finished part, which is equal to that considered in the PBCM (refer to Table 1). The material removal rates (*MRR*) were also assumed to be different to account for realistic cutting conditions: $MRR_1 = 6$ cm³/min (rough machining) and $MRR_2 = 0.6$ cm³/min (fine machining—again equal to that considered in the PBCM).

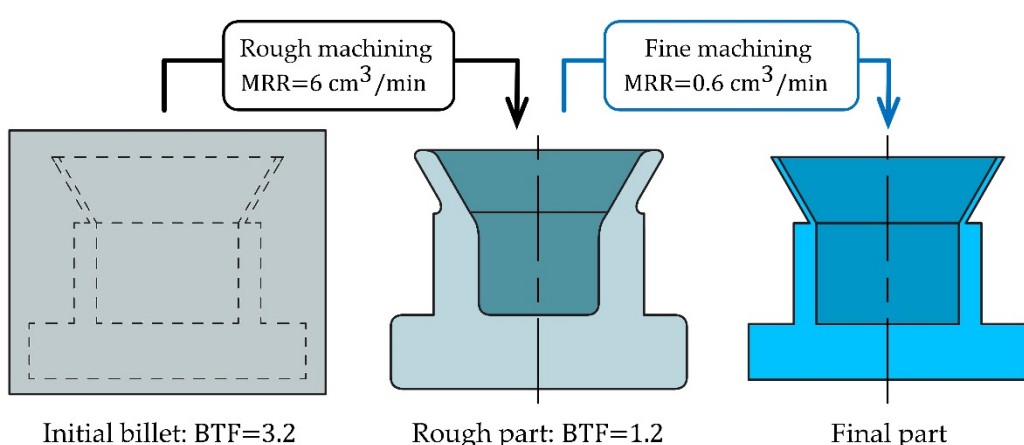

**Figure 8.** Schematic representation of a traditional subtractive approach to obtain the case study part through rough and fine machining tasks. The assumed Buy-to-Fly ratios (*BTF*) and material removal rate (*MRR*) for each task are enclosed as well.

Figure 9a presents a complete comparison of all main individual costs (€ per part) spent when using either a *WAAM* processing chain or a traditional subtractive approach (machining) for the production of the case study part. As seen in Figure 9a, individual part costs related with full machining costs surpass nearly all of those spent on a *WAAM* processing chain. The biggest deviation is denoted on equipment costs (24.28 € vs. 9.82 € per part) because two *CNC* machining centers are required in the traditional subtractive approach. In fact, if only a single *CNC* machining was to be considered for performing both rough and fine machining tasks sequentially, the estimated annual production of 1500 parts would not be attained.

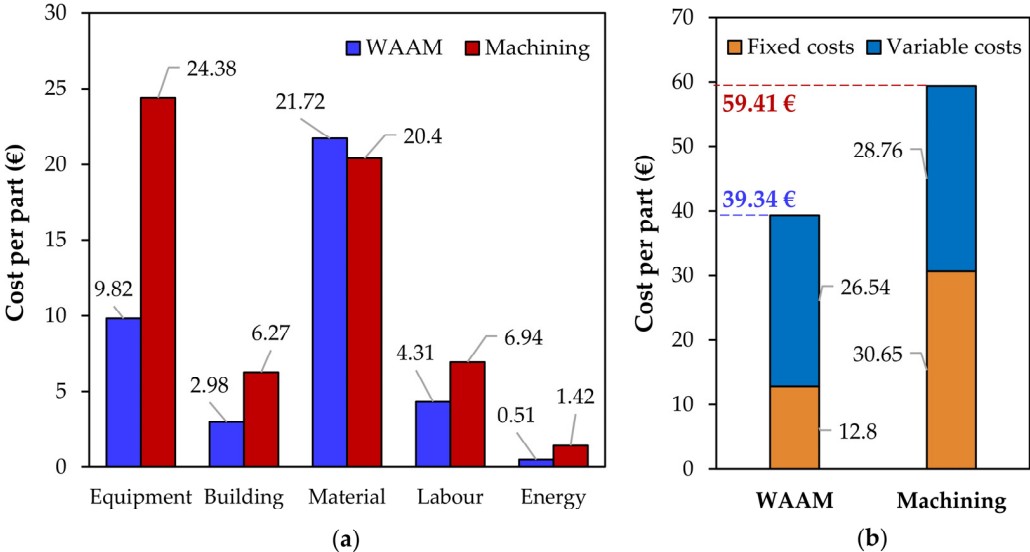

**Figure 9.** Comparison of costs per part between a wire-arc additive manufacturing processing chain (*WAAM*) and a traditional subtractive approach (machining). (**a**) Individual costs. (**b**) Total costs.

Nonetheless, material costs remain are the only higher in *WAAM* when compared to the subtractive approach (20.4 € vs. 21.72 € per part). The reason for this is again due to high acquisition costs of the wire feedstock consumables that are mandatory in *WAAM*. Therefore, even though the initial *BTF* is nearly 3 times higher in the subtractive approach than in *WAAM* (3.2 vs. 1.2), the material price differences related to its format (billets or wire coils) completely countervails the material volume differences. In conclusion, the variable costs will be similar for both approaches, while the fixed costs are those that highlight the expensiveness of using a traditional subtractive approach instead of *WAAM* as shown in Figure 9b. By summing up the main cost drivers previously disclosed in Figure 9b for *WAAM*, the PBCM estimates an overall cost of 39.34 € per part corresponding to a 34% reduction in costs spent for the production of the case study part using a traditional subtractive approach.

For cases where the annual production is higher than 1500 parts, more machines have to be implemented because otherwise the production times would exceed the available machine uptime. Hence, the following cost analysis disclosed in Figure 10 considers the existence of either 1, 10 or 100 fully-dedicated machine sets $N_m$. This new variable $N_m$ serve as a multiplication factor to all fixed costs of the PBCM. For *WAAM*, a machine set $N_m = 1$ implies the utilization of one *WAAM* system (power source + *CNC* gantry) and one machining center. For the traditional subtractive approach, the same machine set $N_m = 1$ now suggests the existence of two machining center: one for rough machining stages and another for fine machining stages.

$$C_{total} = N_m \times C_{fixed} + C_{variable} \tag{16}$$

Figure 10 allows highlighting that the lower cost of *WAAM* is observed for different annual production volumes. This analysis allows denotes the effectiveness of using *WAAM* for producing parts in large batches.

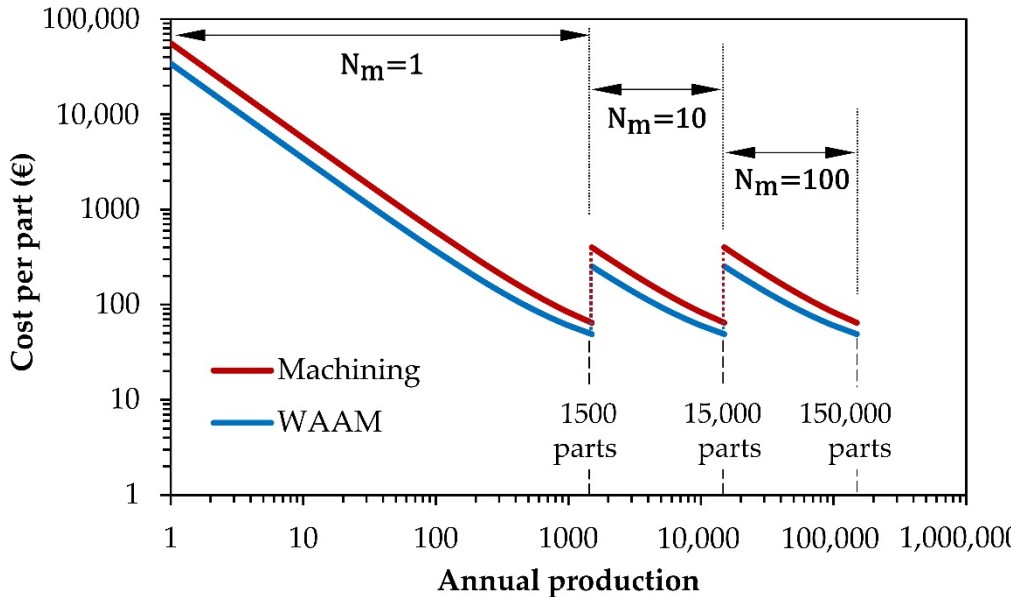

**Figure 10.** Costs per part as function of the annual production for a *WAAM* processing chain and a traditional subtractive approach (machining).

### 3.4. Life Cycle Assessment

The goal and scope of the LCA will follow the same boundaries and comparisons as the cost assessment, considering a cradle to gate approach, the functional unit related to the production of the case study part, and considering its comparison with a traditional subtractive approach (machining). The cradle-to-gate approach considers within the boundaries of the analysis the pre-production and the deposition stage of the part.

Table 2 discloses the inventory used for the LCA analysis. The main environmental drivers, the materials and the energy, were considered in both processes. Regarding consumables, only the argon consumable gas was considered for the *WAAM* processing chain whereas consumables for machining were disregarded since these are negligible in the overall impact [26]. The values were all based on the ones obtained in the case study measures and the remaining values were obtained from a previous study on energy requirements of manufacturing processes [27].

**Table 2.** Inventory used for the Life Cycle Assessment (LCA) analysis.

| Process | Type of Resource | Specification | Amount/Unit |
|---|---|---|---|
| Machining | Materials input | Steel Block 316L | 4 kg |
| | Waste | Steel | 2.75 kg |
| | Energy | Electricity PT mix | 62.4 MJ |
| WAAM | Materials | Wire Steel 316L | 0.902 kg |
| | Waste #1 | Wire Steel 316L | 0.15 kg |
| | Material | Baseplate sheet steel 316L | 0.496 kg |
| | Consumable | Argon Gas | 0.43 kg |
| | Energy | Electricity PT mix | 21.6 MJ |

Table 3 presents the ReCiPe Midpoint impacts values of the part produced by *WAAM* and the impacts of the main environmental drivers (materials and electricity). For the midpoint analysis of *WAAM*, the impacts were assessed in 18 different categories all of them representing different impacts. These are Climate change, Ozone Depletion, Terrestrial Acidification, Freshwater eutrophication, Marine Eutrophication, Human Toxicity,

Photochemical oxidant formation, Terrestrial ecotoxicity, Freshwater ecotoxicity, Marine Ecotoxicity, Ionizing radiation, Agricultural land occupation, Natural land transformation, Water depletion, Metal depletion and Fossil depletion. For a better understanding of each one of these categories it is recommended to see the explanation provided elsewhere [28]. The biggest impact in most categories comes from the material (AISI 316L wire feedstock and baseplate), followed by the electricity and the shielding gas.

**Table 3.** ReCiPe Midpoint analysis for *WAAM*.

| Impact Category | Unit | Total | Wire | Baseplate | Argon | Electricity |
|---|---|---|---|---|---|---|
| Climate change | kg $CO_2$ eq | 7.52 | 3.66 | 2.18 | 0.66 | 1.03 |
| Ozone depletion | kg CFC-11 eq | $4.77 \times 10^{-7}$ | $2.27 \times 10^{-7}$ | $1.31 \times 10^{-7}$ | $4.54 \times 10^{-8}$ | $7.34 \times 10^{-8}$ |
| Terrestrial acidification | kg $SO_2$ eq | $4.59 \times 10^{-2}$ | $2.21 \times 10^{-2}$ | $1.28 \times 10^{-2}$ | $3.91 \times 10^{-3}$ | $7.04 \times 10^{-3}$ |
| Freshwater eutrophication | kg P eq | $2.40 \times 10^{-3}$ | $1.22 \times 10^{-3}$ | $7.12 \times 10^{-4}$ | $2.69 \times 10^{-4}$ | $1.97 \times 10^{-4}$ |
| Marine eutrophication | kg N eq | $9.54 \times 10^{-3}$ | $3.69 \times 10^{-3}$ | $2.07 \times 10^{-3}$ | $1.28 \times 10^{-3}$ | $2.50 \times 10^{-3}$ |
| Human toxicity | kg 1,4-DB eq | 4.03 | 2.31 | 1.31 | 0.22 | 0.19 |
| Photochemical oxidant formation | kg NMVOC | $2.77 \times 10^{-2}$ | $1.43 \times 10^{-2}$ | $8.53 \times 10^{-3}$ | $1.94 \times 10^{-3}$ | $3.01 \times 10^{-3}$ |
| Particulate matter formation | kg PM10 eq | $3.51 \times 10^{-2}$ | $2.01 \times 10^{-2}$ | $1.16 \times 10^{-2}$ | $1.47 \times 10^{-3}$ | $1.86 \times 10^{-3}$ |
| Terrestrial ecotoxicity | kg 1,4-DB eq | $1.26 \times 10^{-3}$ | $7.28 \times 10^{-4}$ | $4.06 \times 10^{-4}$ | $3.87 \times 10^{-5}$ | $8.57 \times 10^{-5}$ |
| Freshwater ecotoxicity | kg 1,4-DB eq | 0.54 | 0.34 | 0.19 | 0.01 | 0.01 |
| Marine ecotoxicity | kg 1,4-DB eq | 0.56 | 0.35 | 0.19 | 0.01 | 0.01 |
| Ionizing radiation | kBq U235 eq | 0.89 | 0.39 | 0.22 | 0.16 | 0.11 |
| Agricultural land occupation | $m^2a$ | 0.62 | 0.32 | 0.18 | 0.03 | 0.10 |
| Urban land occupation | $m^2a$ | 0.12 | 0.07 | 0.04 | 0.00 | 0.01 |
| Natural land transformation | $m^2$ | $7.33 \times 10^{-4}$ | $3.27 \times 10^{-4}$ | $1.94 \times 10^{-4}$ | $7.45 \times 10^{-5}$ | $1.38 \times 10^{-4}$ |
| Water depletion | $m^3$ | 0.16 | −0.02 | −0.01 | 0.20 | 0.00 |
| Metal depletion | kg Fe eq | 18.79 | 12.05 | 6.72 | 0.01 | 0.02 |
| Fossil depletion | kg oil eq | 1.89 | 0.90 | 0.52 | 0.17 | 0.30 |

Finally, the ReCiPe Endpoint method was used for comparing the environmental impact of *WAAM* and machining by means of a single indicator. The results are presented in Figure 11 for each of the endpoint categories, Human Health, Ecosystems and Resources, and as a total (aggregated). Results show the benefit of using an additive manufacturing process (*WAAM*) regarding the material impact. This was expected since the material usage, *BTF* ratio, is much higher when producing a part by traditional subtractive approach than with *WAAM*. The machining equipment is also much more powerful, and therefore requires more energy to operate. The only difference would be the shielding gas, which has a residual impact when compared to those regarding materials and energy. There are no relevant differences proportionally regarding the impact in the categories, as both processes have the main impacts related with material and energy.

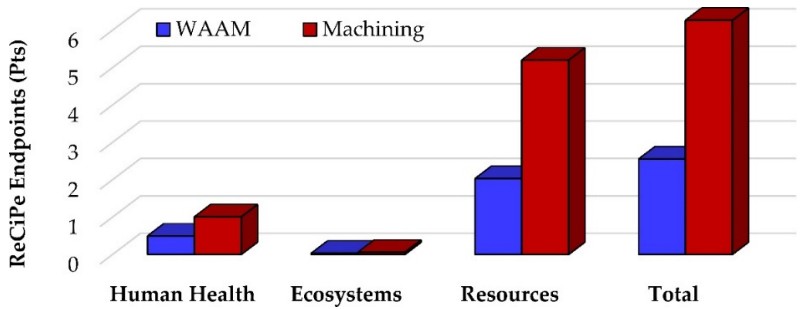

**Figure 11.** ReCiPe endpoint comparison—*WAAM* vs. Machining.

## 4. Conclusions

In this paper, the economic and environmental potential of wire-arc additive manufacturing (*WAAM*) is evaluated firstly with the proposal of a comprehensive process-based cost model (PBCM) and secondly with a life cycle assessment (LCA) through a cradle-to-grave approach. The developments were based on a case study part that allowed comparing the performance of *WAAM* with that of a traditional subtractive approach based on machining from solid. The main conclusions to be drawn from this work are the following:

- The proposed PBCM accounts for the main stages of a typical *WAAM* processing chain including metal deposition tasks and fine machining tasks as well as setup activities related to equipment and materials. This technologically strengthen model allows computing the production costs per part based on the annual production while being also capable of reacting to input variations imposed by industry market dynamics.
- In metal deposition stages, material costs are the key cost driver due to the high wire feedstock price per kilogram whereas for machining, equipment expenses assume the main cost proportions due to their high acquisition costs. By summing all cost parcels together, those spent on material correspond to 55% of the total production costs per part which clearly overcome those spent on equipment use.
- The sensitivity analysis fully corroborates the former conclusion and also allows underlining the possibility of variating the machine uptime for the purpose of leveraging production costs.
- The proposed PBCM estimates a 34% production cost reduction when replacing a traditional subtractive approach with a *WAAM* processing chain. This result can also be extended for large production batches depending on the available number of fully-dedicated machine sets.
- The LCA methodology (ReCiPe2016 method in SimaPro software and the ecoinvent 3.7 database) shows the environmental benefits of *WAAM* when compared with traditional machining due to a much lower material waste. Similar benefits were also found regarding energy consumptions due to the machining equipment being more energy intensive than that of *WAAM*.
- A further extension of this work will be placed on evaluating the economic viability and environmental friendliness on the hybridization of *WAAM* with other manufacturing technologies. For this purpose, the integration of *WAAM*, machining and metal forming operations is among one of the strategies that can be capable of effectively transferring metal additive manufacturing to newer application fields, namely those that require medium to large batch production batches of metallic parts.

**Author Contributions:** Conceptualization, I.R. and C.M.A.S.; methodology, J.P.M.P., I.R. and C.M.A.S.; software, M.D., B.F. and I.R.; validation, M.D. and J.P.M.P.; data curation, M.D. and B.F.; writing—original draft preparation, J.P.M.P.; writing—review and editing, I.R. and C.M.A.S.; visualization, M.D., B.F. and J.P.M.P.; supervision, I.R. and C.M.A.S.; funding acquisition, C.M.A.S. All authors have read and agreed to the published version of the manuscript.

**Funding:** This research received no external funding.

**Institutional Review Board Statement:** Not applicable.

**Informed Consent Statement:** Not applicable.

**Data Availability Statement:** Data is enclosed along the whole extent of the article.

**Acknowledgments:** The authors want to acknowledge the support provided by Fundação para a Ciência e a Tecnologia of Portugal and IDMEC under LAETA-UIDB/50022/2020.

**Conflicts of Interest:** The authors declare no conflict of interest.

### Nomenclature

| | |
|---|---|
| $\varnothing_w$ | Wire feedstock diameter (mm) |
| $C_{Scrap}$ | Waste income per volume of material (€/cm$^3$) |
| $C_{b,w,g}$ | Acquisition unit costs for wire coils $_{(w)}$, baseplates $_{(b)}$ and gas bottles $_{(g)}$ (€) |
| $C_{building}$ | Building costs (€/part) |
| $C_{cutting}$ | Acquisition costs for the cutting tools (€) |
| $C_{energy}$ | Energy consumption costs (€/part) |
| $C_{equipment}$ | Equipment costs (€/part) |
| $C_{fixed}$ | Fixed costs (€/part) |
| $C_{fixing}$ | Tooling costs spent on holders, fixtures and jigs (€) |
| $C_{labor}$ | Labor costs for operators involved in the *WAAM* chain (€/part) |
| $C_{machinery}$ | Machinery costs (€/part) |
| $C_{maintenance}$ | Maintenance costs (€/part) |
| $C_{mat}$ | Material consumable costs (€/part) |
| $C_{tooling}$ | Tooling costs (€/part) |
| $C_{total}$ | Total production costs (€/part) |
| $C_{variable}$ | Variable costs (€/part) |
| $I_{waste}$ | Waste management incomes (€/part) |
| $M_{Labor}$ | Direct wages (€/h) |
| $M_{energy}$ | Monetary price of energy (€/kWh) |
| $M_{machine}$ | Machine investment (€) |
| $N_b$ | Number of reutilizations for the baseplate |
| $N_m$ | Number of fully-dedicated machine sets |
| $N_{life}$ | Machine life span (years) |
| $Q_{gas}$ | Gas flow rate (l/min) |
| $V_{as-built}$ | Volume for the as-built part (after *WAAM*) (cm$^3$) |
| $V_b^{eff}$ | Effective volume of the baseplate for supporting the as-built part (cm$^3$) |
| $V_{gas}$ | Gas volume (l) |
| $V_{part}$ | Volume for the final part (after machining) (cm$^3$) |
| $m_{w,b}$ | Mass of the wire feedstock $_{(w)}$ and baseplate $_{(b)}$ materials (kg) |
| $t_{w,b\ change}$ | Changeover times for wire coils $_{(w)}$ and gas bottle $_{(b)}$ (min) |
| $t_{WAAM}$ | *WAAM* cycle time (deposition time + cooling time) (min) |
| $t_{cooling}$ | Cooling time (min) |
| $t_{cutting}$ | Actual cutting time (min) |
| $t_{dep}$ | Deposition time (min) |
| $t_l$ | Cutting tool lifetime (min) |
| $t_{machining}$ | Machining cycle time (min) |
| $t_{off}$ | Non-productive machining time (min) |
| $t_{set}$ | Time for setting up the *WAAM* or machining equipment (min) |
| $t_{setup}$ | Total setup time (min) |
| $t_u$ | Machine uptime (hours per year) |
| $\rho_{w,b}$ | Density of the wire feedstock $_{(w)}$ and baseplate $_{(b)}$ materials (g/cm$^3$) |
| $AP$ | Annual production (parts per year) |
| $MA$ | Machinery area (m$^2$) |
| $MRR$ | Material removal rate (0.6 cm$^3$/min) |
| $P$ | Machine power consumption (kW) |
| $RR$ | Rejection Rate (%) |
| $WFS$ | Wire feed speed (m/min) |
| $YRR$ | Yearly rent rate (€/m$^2$) |
| $WD$ | Worker dedication (%) |

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
