# Peer review of "Economic and Environmental Potential of Wire-Arc Additive Manufacturing"

_sustainability, doi:10.3390/su14095197_

Round 1
Reviewer 1 Report
- Regarding the cost model analysis, the authors may know that the cost of WAAM is related to the complexity, size and quantity of the printed objects. Also the type of materials used is another influence factor. Please discuss the unconsidered factors.
- For the comparison between WAAM and machining from solid, the details such as how to get the averaged cost for traditional subtractive approach is not clear.
- How about the properties of the as-fabricated objects by WAAM and traditional method? Are they close?
Author Response
The authors are thankful for the comments and remarks provided from the reviewer that helped improving the manuscript. The answers together with appropriate references to the modifications that were introduced in the revised version of the manuscript are included in what follows.
- Regarding the cost model analysis, the authors may know that the cost of WAAM is related to the complexity, size and quantity of the printed objects. Also the type of materials used is another influence factor. Please discuss the unconsidered factors.
Answer:
The authors agree that part complexity, size and quantity are among the main cost drivers involved in WAAM and play an important role in process decision planning. The effect of part size on the overall production costs was addressed with the inclusion of part volume (before and after machining) and material density in the cost model. The effect of part quantity is directly related to the annual production which is also an input from the model (refer to table 1).
Regarding part complexity, it was assessed by a volume-dependent approach as a function of the processing parameters of WAAM. This allowed the authors to take into consideration their past experience in WAAM parametrization for the purpose of constructing the case study part while also to focus their attention in the main direct and variable cost drivers, which is the primary purpose of the manuscript.
Finally, the effect of the material alloy on costs is split in multiple inputs such as material density, baseplate acquisition costs and wire acquisition costs. The latter two variables require the user to input their respective values because they can vary from company to company (the material supplier). Furthermore, changing the material may require using different processing parameters according to the application. Therefore, the calculated production costs are expected to change as well.
The above information was included in the revised version of the manuscript.
- For the comparison between WAAM and machining from solid, the details such as how to get the averaged cost for traditional subtractive approach is not clear.
Answer:
The cost estimations for the traditional subtractive approach (machined from solid) are mainly based upon the considerations presented in subsection 2.2 for machining tasks carried out in WAAM as-built parts (equations 3, 4, 5, 6 ,13, 14 and 15 of the manuscript). The main differences are shown in the beginning of subsection 3.3 which propose utilizing two different machining stages (rough machining and fine machining). Both these stages are usually carried out sequentially in traditional machining approaches because they allow decreasing machining times without compromising the final part quality standards in terms of surface conditions and geometry precision.
Given the material removal rates (MRR) and Buy-to-Fly ratios (BTF) of each stage that allow estimating their machining times, added with the technical characteristics of the machining CNC centre, the authors were able to compute the production costs of the traditional subtractive approach regarding Equipment, Building Material, Labor and Energy (refer to Figure 9). Hence, these costs do not represent average values but instead the calculated production costs per part.
The aforementioned points were further clarified in subsection 3.3 of the revised version of the manuscript.
- How about the properties of the as-fabricated objects by WAAM and traditional method? Are they close?
Answer:
The material properties of the WAAM parts were not analysed because the authors considered that to be out of the scope of the manuscript. However, the authors have already addressed the mechanical and metallurgical characteristics of WAAM-based stainless steel AISI 316L in comparison to those of commercial raw materials of the same metal alloy. In these works, the deposited material was found to have an irregular dendritic based microstructure attributed by the heating-cooling cycles taking place during layer-by-layer construction. This is a common issue in most metals when processed by metal additive manufacturing processes that leads to materials having an anisotropic behaviour that is highly related to the building direction.
Still, even though the formability limits of WAAM-based AISI 316L stainless steel were found to be smaller than their commercial reference, its ductility is still high enough to be ensure a proper performance without risk of failure.
The following reference and more information were included in the revised version of the manuscript referring to the properties of WAAM-based AISI 316L stainless steel:
J. P. Pragana, I. M. Bragança, L. Reis, C. M. Silva and P. A. Martins, “Formability of wire-arc deposited AISI 316L sheets for hybrid additive manufacturing applications,” Proceedings of the Institution of Mechanical Engineers, Part L: Journal of Materials: Design and Applications, vol 235, 2839-2850, 2021, doi: 10.1177/14644207211037033
Thank you very much once again for the revision of the manuscript.
Reviewer 2 Report
accept
Author Response
Thank you very much for the revision of the manuscript.
Reviewer 3 Report
The presented topic "Economic and environmental potential of wire-arc additive 2 manufacturing" is interesting for readers. The paper is well written but has plenty of room for improvement:
1. Introduction section - specify clearer goals of the work. What are the novelties of the paper in relation to the cited references, ie what does the paper bring new in comparison with similar papers in the literature.
2. Fig. 3, give a more detailed description.
3. Section 2, rows 106-107 "The working principles of WAAM involve layer-by-layer deposition in which an electric arc is used as a thermal energy source for melting the wire feedstock. The way heat is transferred from the wire to the workpiece is very important for WAAM as it has direct effects on deformations and residual stresses. Here it would be good to refer the readers and cite the following recent references (https://doi.org/10.1007/s10973-019-09231-3 and https://doi.org /10.1007/s00170-021-08037-8).
4. Conclusion section, here it would be good to write conclusions in bullet form to better highlight the main contributions of the paper).
5. Add Nomenclature
6. Future work, if any.
Author Response
The authors are thankful for the comments and remarks provided from the reviewer that helped improving the manuscript. The answers together with appropriate references to the modifications that were introduced in the revised version of the manuscript are included in what follows.
The presented topic "Economic and environmental potential of wire-arc additive 2 manufacturing" is interesting for readers. The paper is well written but has plenty of room for improvement:
Answer:
Thank you very much for the comments and for the revision of the manuscript.
- Introduction section - specify clearer goals of the work. What are the novelties of the paper in relation to the cited references, ie what does the paper bring new in comparison with similar papers in the literature.
Answer:
The main contribution of the manuscript is to provide a combined economic and environmental assessment of WAAM in comparison with other conventional manufacturing technologies such as machining. As far as the authors are concerned, this combined study has not been yet attempted and allows highlighting the economic viability and environmental friendliness of WAAM that can in fact be settled as an efficient and sustainable alternative to conventional manufacturing technologies.
Regarding only cost modelling analysis, a considerable amount of research works can be found in the literature addressing the cost performance of additive manufacturing processes such as Direct Metal Laser Sintering, Electron Beam Melting, Fused Deposition Modelling, among others, but only very few are focused on WAAM. These few works were those of Martina and Williams [17] and of Cunningham et al. [18] that although being considered by the authors as milestones in what comes to analysing production costs in WAAM, they do not consider certain variables that can have a key role in cost evaluation. Examples of such are non-production tasks, modifications in acquisition costs of consumables, variable labor, among others. For these reasons, the process-based cost model proposed by the authors can be seen as a technologically strengthen model capable of reacting to most industry market dynamics while also being supported by the environmental assessment of WAAM.
The following reference and more information were included in the revised version of the manuscript to further clarify the novelty of the proposed work:
G. Costabile, M. Fera, F. Fruggiero, A. Lambiase and D. Pham, “Cost models of additive manufacturing: A literature review,” International Journal of Industrial Engineering Computations, vol 8, 263-283, 2017, doi: 10.5267/j.ijiec.2016.9.001
- Figure 3, give a more detailed description.
Answer:
The caption of Figure 3 was further improved in the revised version of the manuscript in order to fully describe the measured power consumptions from the WAAM system (gas metal arc welding power source + CNC gantry).
- Section 2, rows 106-107 "The working principles of WAAM involve layer-by-layer deposition in which an electric arc is used as a thermal energy source for melting the wire feedstock. The way heat is transferred from the wire to the workpiece is very important for WAAM as it has direct effects on deformations and residual stresses. Here it would be good to refer the readers and cite the following recent references (https://doi.org/10.1007/s10973-019-09231-3 and https://doi.org /10.1007/s00170-021-08037-8).
Answer:
The following reference was added in Section 2 of the revised version of the manuscript:
- Wang, S. Zimmer-Chevret, F. Léonard, G. Abba, “Improvement strategy for the geometric accuracy of bead’s beginning and end parts in wire-arc additive manufacturing (WAAM),” International Journal of Advanced Manufacturing Technology, vol 118, 2139-2151, 2022, doi: 10.1007/s00170-021-08037-8
- Conclusion section, here it would be good to write conclusions in bullet form to better highlight the main contributions of the paper.
Answer:
Section 4 of the revised version of the manuscript was rewritten in bullet form for further emphasising its main findings and contributions.
- Add Nomenclature
Answer:
A new section was added in the revised version of the manuscript (in-between the ‘Conclusions’ and ‘References’ sections) that provides a complete nomenclature list of the symbols disclosed along the manuscript.
- Future work, if any.
Answer:
The authors intend to further extend the proposed work in evaluating the economic viability and environmental friendliness on the hybridization of WAAM with other manufacturing technologies. Regarding this research line, the integration of WAAM, machining and metal forming operations is among one of the strategies that can be capable of effectively transferring metal additive manufacturing to newer application fields, namely those that require medium to large batch production batches of metallic parts.
This information is included in the last paragraph of Section 4 of the revised version of the manuscript.
Thank you very much once again for the revision of the manuscript.
Reviewer 4 Report
Additive manufacturing (AM) is a group of technologies that create objects by adding material layer upon layer, in precise geometric shapes. They are amongst the most disruptive technologies nowadays, potentially changing value chains from the design process to the end-of-life, providing significant advantages over traditional manufacturing processes in terms of flexibility in design and production and waste minimization. Reviewed manuscript is focused on an emerging AM process known as Wire-Arc Additive Manufacturing (WAAM) to assess its potential for further applications involving metallic costumer-oriented parts.
Manuscript entitled "Economic and environmental potential of wire-arc additive manufacturing" of interest to "Sustainability" journal. Presented work opens up prospects in this field of knowledge.
It should be noted that this manuscript reflects the development of research presented in [2]: Ribeiro et al., “Framework for life cycle sustainability assessment of additive manufacturing,” Sustain., vol. 12, no. 3, 2020, doi: 538 10.3390/su12030929.
Weakness and methodological inaccuracies are not detected. Reviewed manuscript is clear, relevant for the field and presented in a well-structured manner. The cited references are current (including within the last 5 years). The manuscript is scientifically sound and the manuscript’s results are reproducible based on the details given in the "Methodologies" section (p. 3-7). The figures and tables are appropriate, they properly show the data. So, they easy to interpret and understand. Thus, the data 100% interpreted appropriately and consistently throughout the manuscript.
Results allow highlighting the environmental benefits of WAAM when compared with traditional machining as a result of the lower material waste in the process. Furthermore, the machining equipment is more energy intensive than the WAAM equipment used.
It is worth agreeing with the authors that further developments and optimizations of process variables and equipment will allow this technology to mature into tackling novel production challenges in a time and cost-effective manner.
Overall Recommendation: Accept in present form.
Author Response
Thank you very much for the detailed comments and for the revision of the manuscript.
Round 2
Reviewer 3 Report
*